# THIRD-PERSON IMITATION LEARNING

**Bradly C. Stadie**[1,2]**, Pieter Abbeel**[1,3]**, Ilya Sutskever**[1]
[1] OpenAI
[2] UC Berkeley, Department of Statistics
[3] UC Berkeley, Departments of EECS and ICSI
{bstadie, pieter, ilyasu,}@openai.com

## ABSTRACT

Reinforcement learning (RL) makes it possible to train agents capable of achieving sophisticated goals in complex and uncertain environments. A key difficulty in reinforcement learning is specifying a reward function for the agent to optimize. Traditionally, imitation learning in RL has been used to overcome this problem. Unfortunately, hitherto imitation learning methods tend to require that demonstrations are supplied in the *first-person*: the agent is provided with a sequence of states and a specification of the actions that it should have taken. While powerful, this kind of imitation learning is limited by the relatively hard problem of collecting first-person demonstrations. Humans address this problem by learning from *third-person* demonstrations: they observe *other humans* perform tasks, infer the task, and accomplish the same task themselves.

In this paper, we present a method for *unsupervised* third-person imitation learning. Here third-person refers to training an agent to correctly achieve a simple goal in a simple environment when it is provided a demonstration of a teacher achieving the same goal but from a different viewpoint; and unsupervised refers to the fact that the agent receives only these third-person demonstrations, and is *not* provided a correspondence between teacher states and student states. Our methods primary insight is that recent advances from domain confusion can be utilized to yield domain agnostic features which are crucial during the training process. To validate our approach, we report successful experiments on learning from third-person demonstrations in a pointmass domain, a reacher domain, and inverted pendulum.

## 1 INTRODUCTION

Reinforcement learning (RL) is a framework for training agents to maximize rewards in large, unknown, stochastic environments. In recent years, combining techniques from deep learning with reinforcement learning has yielded a string of successful applications in game playing and robotics Mnih et al. (2015; 2016); Schulman et al. (2015a); Levine et al. (2016). These successful applications, and the speed at which the abilities of RL algorithms have been increasing, makes it an exciting area of research with significant potential for future applications.

One of the major weaknesses of RL is the need to manually specify a reward function. For each task we wish our agent to accomplish, we must provide it with a reward function whose maximizer will precisely recover the desired behavior. This weakness is addressed by the field of Inverse Reinforcement Learning (IRL). Given a set of expert trajectories, IRL algorithms produce a reward function under which these the expert trajectories enjoy the property of optimality. Recently, there has been a significant amount of work on IRL, and current algorithms can infer a reward function from a very modest number of demonstrations (e.g,. Abbeel & Ng (2004); Ratliff et al. (2006); Ziebart et al. (2008); Levine et al. (2011); Ho & Ermon (2016); Finn et al. (2016)).

While IRL algorithms are appealing, they impose the somewhat unrealistic requirement that the demonstrations should be provided from the *first-person* point of view with respect to the agent. Human beings learn to imitate entirely from third-person demonstrations – i.e., by observing other humans achieve goals. Indeed, in many situations, first-person demonstrations are outright impossible to obtain. Meanwhile, third-person demonstrations are often relatively easy to obtain.

The goal of this paper is to develop an algorithm for third-person imitation learning. Future advancements in this class of algorithms would significantly improve the state of robotics, because it will enable people to easily teach robots news skills and abilities. Importantly, we want our algorithm to be *unsupervised*: it should be able to observe another agent perform a task, infer that there is an underlying correspondence to itself, and find a way to accomplish the same task.

We offer an approach to this problem by borrowing ideas from domain confusion Tzeng et al. (2014) and generative adversarial networks (GANs) Goodfellow et al. (2014). The high-level idea is to introduce an optimizer under which we can recover both a domain-agnostic representation of the agent's observations, and a cost function which utilizes this domain-agnostic representation to capture the essence of expert trajectories. We formulate this as a third-person RL-GAN problem, and our solution builds on the first-person RL-GAN formulation by Ho & Ermon (2016).

Surprisingly, we find that this simple approach has been able to solve the problems that are presented in this paper (illustrated in Figure 1), even though the student's observations are related in a complicated way to the teacher's demonstrations (given that the observations and the demonstrations are pixel-level). As techniques for training GANs become more stable and capable, we expect our algorithm to be able to infer solve harder third-person imitation tasks without any direct supervision.

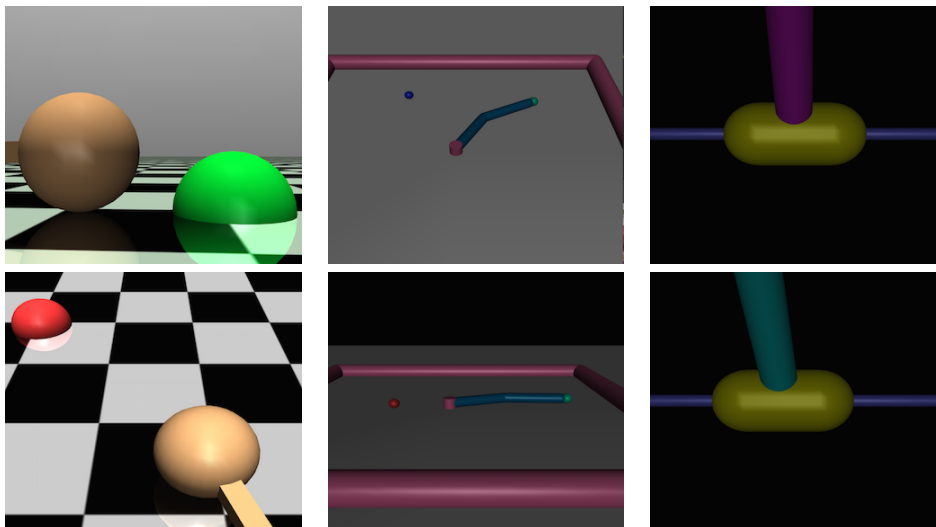

Figure 1: From left to right, the three domains we consider in this paper: pointmass, reacher, and pendulum. Top-row is the third-person view of a teacher demonstration. Bottom row is the agent's view in their version of the environment. For the point and reacher environments, the camera angles differ by approximately 40 degrees. For the pendulum environment, the color of the pole differs.

## 2 RELATED WORK

Imitation learning (also learning from demonstrations or programming by demonstration) considers the problem of acquiring skills from observing demonstrations. Imitation learning has a long history, with several good survey articles, including (Schaal, 1999; Calinon, 2009; Argall et al., 2009). Two main lines of work within imitation learning are: 1) behavioral cloning, where the demonstrations are used to directly learn a mapping from observations to actions using supervised learning, potentially with interleaving learning and data collection (e.g., Pomerleau (1989); Ross et al. (2011)). 2) Inverse reinforcement learning (Ng et al., 2000), where a reward function is estimated that explains the demonstrations as (near) optimal behavior. This reward function could be represented as nearness to a trajectory (Calinon et al., 2007; Abbeel et al., 2010), as a weighted combination of features (Abbeel & Ng, 2004; Ratliff et al., 2006; Ramachandran & Amir, 2007; Ziebart et al., 2008; Boularias et al., 2011; Kalakrishnan et al., 2013; Doerr et al., 2015), or could also involve feature learning (Ratliff et al., 2007; Levine et al., 2011; Wulfmeier et al., 2015; Finn et al., 2016; Ho & Ermon, 2016).

This past work, however, is not directly applicable to the *third* person imitation learning setting. In third-person imitation learning, the observations and actions obtained from the demonstrations are *not* the same as what the imitator agent will be faced with. A typical scenario would be: the imitator agent watches a human perform a demonstration, and then has to execute that same task. As discussed in Nehaniv & Dautenhahn (2001) the "what and how to imitate" questions become significantly more challenging in this setting. To directly apply existing behavioral cloning or inverse reinforcement learning techniques would require knowledge of a mapping between observations and actions in the demonstrator space to observations and actions in the imitator space. Such a mapping is often difficult to obtain, and it typically relies on providing feature representations that captures the invariance between both environments Carpenter et al. (2002); Shon et al. (2005); Calinon et al. (2007); Nehaniv (2007); Gioioso et al. (2013); Gupta et al. (2016). Contrary to prior work, we consider third-person imitation learning from raw sensory data, where no such features are made available.

The most closely related work to ours is by Finn et al. (2016); Ho & Ermon (2016); Wulfmeier et al. (2015), who also consider inverse reinforcement learning directly from raw sensory data. However, the applicability of their approaches is limited to the first-person setting. Indeed, matching raw sensory observations is impossible in the 3rd person setting.

Our work also closely builds on advances in generative adversarial networks Goodfellow et al. (2014), which are very closely related to imitation learning as explained in Finn et al. (2016); Ho & Ermon (2016). In our optimization formulation, we apply the gradient flipping technique from Ganin & Lempitsky (2014).

The problem of adapting what is learned in one domain to another domain has been studied extensively in computer vision in the supervised learning setting Yang et al. (2007); Mansour et al. (2009); Kulis et al. (2011); Aytar & Zisserman (2011); Duan et al. (2012); Hoffman et al. (2013); Long & Wang (2015). It has also been shown that features trained in one domain can often be relevant to other domains Donahue et al. (2014). The work most closely related to ours is Tzeng et al. (2014; 2015), who also consider an explicit domain confusion loss, forcing trained classifiers to rely on features that don't allow to distinguish between two domains. This work in turn relates to earlier work by Bromley et al. (1993); Chopra et al. (2005), which also considers supervised training of deep feature embeddings.

Our approach to third-person imitation learning relies on reinforcement learning from raw sensory data in the imitator domain. Several recent advances in deep reinforcement learning have made this practical, including Deep Q-Networks (Mnih et al., 2015), Trust Region Policy Optimization (Schulman et al., 2015a), A3C Mnih et al. (2016), and Generalized Advantage Estimation (Schulman et al., 2015b). Our approach uses Trust Region Policy Optimization.

## 3 BACKGROUND AND PRELIMINARIES

A discrete-time finite-horizon discounted Markov decision process (MDP) is represented by a tuple $M = (\mathcal{S}, \mathcal{A}, \mathcal{P}, r, \rho_0, \gamma, T)$, in which $\mathcal{S}$ is a state set, $\mathcal{A}$ an action set, $\mathcal{P} : \mathcal{S} \times \mathcal{A} \times \mathcal{S} \to \mathbb{R}_+$ a transition probability distribution, $r : \mathcal{S} \times \mathcal{A} \to \mathbb{R}$ a reward function, $\rho_0 : \mathcal{S} \to \mathbb{R}_+$ an initial state distribution, $\gamma \in [0, 1]$ a discount factor, and $T$ the horizon.

In the reinforcement learning setting, the goal is to find a policy $\pi_\theta : \mathcal{S} \times \mathcal{A} \to \mathbb{R}_+$ parametrized by $\theta$ that maximizes the expected discounted sum of rewards incurred, $\eta(\pi_\theta) = \mathbb{E}_{\pi_\theta}[\sum_{t=0}^{T} \gamma^t c(s_t)]$, where $s_0 \sim \rho_0(s_0)$, $a_t \sim \pi_\theta(a_t|s_t)$, and $s_{t+1} \sim \mathcal{P}(s_{t+1}|s_t, a_t)$.

In the (first-person) imitation learning setting, we are not given the reward function. Instead we are given traces (i.e., sequences of states traversed) by an expert who acts according to an unknown policy $\pi_E$. The goal is to find a policy $\pi_\theta$ that performs as well as the expert against the unknown reward function. It was shown in Abbeel & Ng (2004) that this can be achieved through inverse reinforcement learning by finding a policy $\pi_\theta$ that matches the expert's empirical expectation over discounted sum of all features that might contribute to the reward function. The work by Ho & Ermon (2016) generalizes this to the setting when no features are provided as follows: Find a policy $\pi_\theta$ that makes it impossible for a discriminator (in their work a deep neural net) to distinguish states visited by the expert from states visited by the imitator agent. This can be formalized as follows:

$$\max_{\pi_\theta} \min_{\mathcal{D}_R} \quad - \mathbb{E}_{\pi_\theta}[\log \mathcal{D}_R(s)] - \mathbb{E}_{\pi_E}[\log(1 - \mathcal{D}_R(s))] \tag{1}$$

Here, the expectations are over the states experienced by the policy of the imitator agent, $\pi_\theta$, and by the policy of the expert, $\pi_E$, respectively. $\mathcal{D}_R$ is the discriminator, which outputs the probability of a state having originated from a trace from the imitator policy $\pi_\theta$. If the discriminator is perfectly able to distinguish which policy originated state-action pairs, then $\mathcal{D}_R$ will consistently output a probability of 1 in the first term, and a probability of 0 in the second term, making the objective its lowest possible value of zero. It is the role of the imitator agent $\pi_\theta$ to find a policy that makes it difficult for the discriminator to make that distinction. The desired equilibrium has the imitator agent making it impractical for the discriminator to distinguish, hence forcing the discriminator to assign probability 0.5 in all cases. Ho & Ermon (2016) present a practical approach for solving this type of game when representing both $\pi_\theta$ and $\mathcal{D}_R$ as deep neural networks. Their approach repeatedly performs gradient updates on each of them. Concretely, for a current policy $\pi_\theta$ traces can be collected, which together with the expert traces form a data-set on which $\mathcal{D}_R$ can be trained with supervised learning minimizing the negative log-likelihood (in practice only performing a modest number of updates). For a fixed $\mathcal{D}_R$, this is a policy optimization problem where $-\log \mathcal{D}_R(s, a)$ is the reward, and policy gradients can be computed from those same traces. Their approach uses trust region policy optimization (Schulman et al., 2015a) to update the imitator policy $\pi_\theta$ from those gradients.

In our work we will have more terms in the objective, so for compactness of notation, we will realize the discriminative minimization from Eqn. (1) as follows:

$$\max_{\pi_\theta} \min_{\mathcal{D}_R} \mathcal{L}_R = \sum_i CE(\mathcal{D}_R(s_i), c_{\ell_i}) \tag{2}$$

Where $s_i$ is state $i$, $c_{\ell_i}$ is the correct class label (was the state $s_i$ obtained from an expert vs. from a non-expert), and $CE$ is the standard cross entropy loss.

## 4 A FORMAL DEFINITION OF THE THIRD-PERSON IMITATION LEARNING PROBLEM

Formally, the third-person imitation learning problem can be stated as follows. Suppose we are given two Markov Decision Processes $M_{\pi_E}$ and $M_{\pi_\theta}$. Suppose further there exists a set of traces $\rho = \{(s_1, \ldots, s_n)\}_{i=0}^n$ which were generated under a policy $\pi_E$ acting optimally under some unknown reward $R_{\pi_E}$. In third-person imitation learning, one attempts to recover by proxy through $\rho$ a policy $\pi_\theta = f(\rho)$ which acts optimally with respect to $R_{\pi_\theta}$.

## 5 A THIRD-PERSON IMITATION LEARNING ALGORITHM

### 5.1 GAME FORMULATION

In this section, we discuss a simple algorithm for third-person imitation learning. This algorithm is able to successfully discriminate between expert and novice policies, even when the policies are executed under different environments. Subsequently, this discrimination signal can be used to train expert policies in new domains via RL by training the novice policy to fool the discriminator, thus forcing it to match the expert policy.

In third-person learning, observations are more typically available rather than direct state access, so going forward we will work with observations $o_t$ instead of states $s_t$ as representing the expert traces. The top row of Figure 8 illustrates what these observations are like in our experiments.

We begin by recalling that in the algorithm proposed by Ho & Ermon (2016) the loss in Equation 2 is utilized to train a discriminator $\mathcal{D}_R$ capable of distinguishing expert vs non-expert policies. Unfortunately, (2) will likely fail in cases when the expert and non-expert act in different environments, since $\mathcal{D}_R$ will quickly learn these differences and use them as a strong classification signal.

To handle the third-person setting, where expert and novice are in different environments, we consider that $\mathcal{D}_R$ works by first extracting features from $o_t$, and then using these features to make a

classification. Suppose then that we partition $\mathcal{D}_R$ into a feature extractor $\mathcal{D}_F$ and the actual classifier which assigns probabilities to the outputs of $D_F$. Overloading notation, we will refer to the classifier as $\mathcal{D}_R$ going forward. For example, in case of a deep neural net representation, $\mathcal{D}_F$ would correspond to the earlier layers, and $\mathcal{D}_R$ to the later layers. The problem is then to ensure that $D_F$ contains no information regarding the rollout's domain label $d_\ell$ (i.e., expert vs. novice domain). This can be realized as

$$\max_{\pi_\theta} \min \mathcal{L}_R = \sum_i CE(\mathcal{D}_R(\mathcal{D}_F(o_i)), c_{\ell_i})$$

$$\text{s.t. } \mathrm{MI}(D_F(o_i); d_l) = 0$$

Where MI is mutual information and hence we have abused notation by using $\mathcal{D}_R$, $D_F$, and $d_\ell$ to mean the classifier, feature extractor, and the domain label respectively as well as distributions over these objects.

The mutual information term can be instantiated by introducing another classifier $\mathcal{D}_D$, which takes features produced by $D_F$ and outputs the probability that those features were produced by in the expert vs. non-expert environment. (See Bridle et al. (1992); Barber & Agakov (2005); Krause et al. (2010); Chen et al. (2016) for further discussion on instantiating the information term by introducing another classifier.) If $\sigma_i = D_F(o_i)$, then the problem can be written as

$$\max_{\pi_\theta} \min_{\mathcal{D}_R} \max_{\mathcal{D}_D} \mathcal{L}_R + \mathcal{L}_D = \sum_i CE(\mathcal{D}_R(\sigma_i), c_{\ell_i}) + CE(\mathcal{D}_D(\sigma_i), d_{\ell_i}) \tag{3}$$

In words, we wish to minimize class loss while maximizing domain confusion.

Often, it can be difficult for even humans to judge a static image as expert vs. non-expert because it does not convey any information about the environmental change affected by the agent's actions. For example, if a pointmass is attempting to move to a target location and starts far away from its goal state, it can be difficult to judge if the policy itself is bad or the initialization was simply unlucky. In response to this difficulty, we give $\mathcal{D}_R$ access to not only the image at time $t$, but also at some future time $t + n$. Define $\sigma_t = D_F(o_t)$ and $\sigma_{t+n} = D_F(o_{t+n})$. The classifier then makes a prediction $\mathcal{D}_R(\sigma_t, \sigma_{t+n}) = \hat{c}_\ell$.

This renders the following formulation:

$$\max_{\pi_\theta} \min_{\mathcal{D}_R} \max_{\mathcal{D}_D} \mathcal{L}_R + \mathcal{L}_D = \sum_i CE(\mathcal{D}_R(\sigma_i, \sigma_{i+n}), c_{\ell_i}) + CE(\mathcal{D}_D(\sigma_i), d_{\ell_i}) \tag{4}$$

Note we also want to optimize over $\mathcal{D}_F$, the feature extractor, but it feeds both into $\mathcal{D}_R$ and into $\mathcal{D}_D$, which are competing (hidden under $\sigma$), which we will address now.

To deal with the competition over $\mathcal{D}_F$, we introduce a function $\mathcal{G}$ that acts as the identity when moving forward through a directed acyclic graph and flips the sign when backpropagating through the graph. This technique has enjoyed recent success in computer vision. See, for example, (Ganin & Lempitsky, 2014). With this trick, the problem reduces to its final form

$$\max_{\pi_\theta} \min_{\mathcal{D}_R, \mathcal{D}_D, \mathcal{D}_F} \mathcal{L}_R + \mathcal{L}_D = \sum_i CE(\mathcal{D}_R(\sigma_i, \sigma_{i+n}), c_{\ell_i}) + \lambda\, CE(\mathcal{D}_D(\mathcal{G}(\sigma_i)), d_{\ell_i}) \tag{5}$$

In Equation (5), we flip the gradient's sign during backpropagation of $D_F$ with respect to the domain classification loss. This corresponds to stochastic gradient ascent away from features that are useful for domain classification, thus ensuring that $D_F$ produces domain agnostic features. Equation 5 can be solved efficiently with stochastic gradient descent. Here $\lambda$ is a hyperparameter that determines the trade-off made between the objectives that are competing over $\mathcal{D}_F$.

To ensure sufficient signal for discrimination between expert and non-expert, we collect third-person demonstrations in the expert domain from both an expert and from a non-expert.

Our complete formulation is graphically summarized in Figure 2.

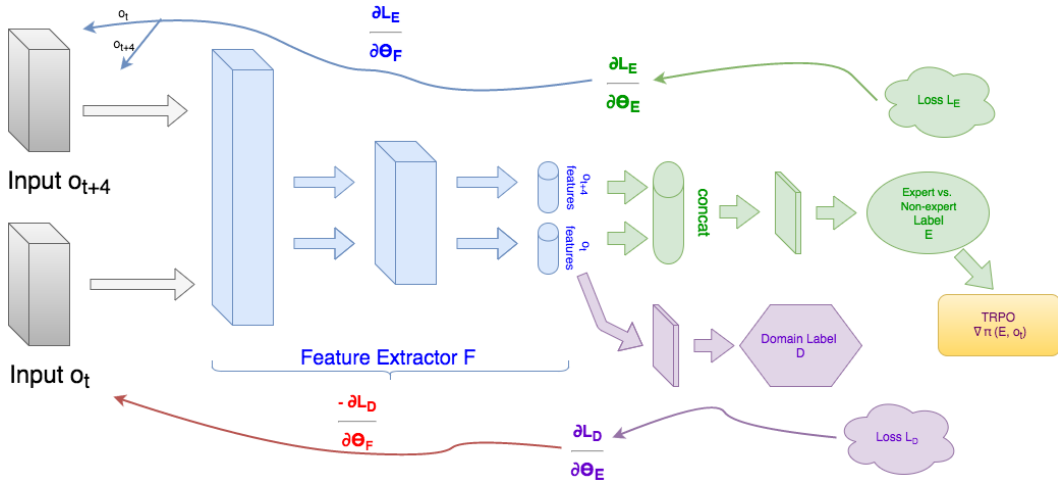

Figure 2: Architecture diagram for third-person imitation learning. Images at time $t$ and $t+4$ are sent through a feature extractor to obtain $F(o_t)$ and $F(o_{t+4})$. Subsequently, these feature vectors are reused in two places. First, they are concatenated and used to predict whether the samples are drawn from expert or non-expert trajectories. Second, $F(o_t)$ is utilized to predict a domain label (expert vs. novice domain). During backpropagation, the sign on the domain loss $L_D$ is flipped to destroy information that was useful for distinguishing the two domains. This ensures that the feature extractor $F$ is domain agnostic. Finally, the classes probabilities that were computed using this domain-agnostic feature vector are utilized as a cost signal in TRPO; which is subsequently utilized to train the novice policy to take expert-like actions and collect further rollouts.

## 5.2 ALGORITHM

To solve the game formulation in Equation (5), we perform alternating (partial) optimization over the policy $\pi_\theta$ and the reward function and domain confusion encoded through $\mathcal{D}_R, \mathcal{D}_D, \mathcal{D}_F$.

The optimization over $\mathcal{D}_R, \mathcal{D}_D, \mathcal{D}_F$ is done through stochastic gradient descent with ADAM Kingma & Ba (2014).

Our generator ($\pi_\theta$) step is similar to the generator step in the algorithm by (Ho & Ermon, 2016). We simply use $-\log \mathcal{D}_R$ as the reward. Using policy gradient methods (TRPO), we train the generator to minimize this cost and thus push the policy further towards replicating expert behavior. Once the generator step is done, we start again with the discriminator step. The entire process is summarized in algorithm 1.

## 6 EXPERIMENTS

We seek to answer the following questions through experiments:

1. Is it possible to solve the third-person imitation learning problem in simple settings? I.e., given a collection of expert image-based rollouts in one domain, is it possible to train a policy in a different domain that replicates the essence of the original behavior?

2. Does the algorithm we propose benefit from both domain confusion and velocity?

3. How sensitive is our proposed algorithm to the selection of hyper-parameters used in deployment?

4. How sensitive is our proposed algorithm to changes in camera angle?

5. How does our method compare against some reasonable baselines?

---

**Algorithm 1** A third-person imitation learning algorithm.
---
1: Let CE be the standard cross entropy loss.
2: Let $\mathcal{G}$ be a function that flips the gradient sign during backpropogation and acts as the identity map otherwise.
3: Initialize two domains, $E$ and $N$ for the expert and novice.
4: Initialize a memory bank $\Omega$ of expert success and of failure in domain $E$. Each trajectory $\omega \in \Omega$ comprises a rollout of images $o = o_1, \ldots, o_t, \ldots o_n$, a class label $c_\ell$, and a domain label $d_\ell$.
5: Initialize $\mathcal{D} = \mathcal{D}_F, \mathcal{D}_R, \mathcal{D}_D$, a domain invariant discriminator.
6: Initialize a novice policy $\pi_\theta$.
7: Initialize numiters, the number of inner policy optimization iterations we wish to run.
8: **for** iter in numiters **do**
9:     Sample a set of successes and failures $\omega_E$ from $\Omega$.
10:     Collect on policy samples $\omega_N$
11:     Set $\omega = \omega_E \cup \omega_N$.
12:     Shuffle $\omega$
13:     **for** $o, c_\ell, d_\ell$ in $\omega$ **do**
14:        **for** $o_t$ in $o$ **do**
15:            $\sigma_t = \mathcal{D}_F(o_t)$
16:            $\sigma_{t+4} = \mathcal{D}_F(o_{t+4})$
17:            $\mathcal{L}_R = CE(\mathcal{D}_R(\sigma_t, \sigma_{t+4}), c_\ell)$
18:            $\mathcal{L}_d = CE(\mathcal{D}_D(\mathcal{G}(\sigma_t)), d_\ell)$
19:            $\mathcal{L} = \lambda \cdot \mathcal{L}_d + \mathcal{L}_R$ [1]
20:            minimize $\mathcal{L}$ with ADAM.
21:        **end for**
22:     **end for**
23:     Collect on policy samples $\omega_N$ from $\pi_\theta$.
24:     **for** $\omega$ in $\omega_N$ **do**
25:        **for** $\omega_t$ in $\omega$ **do**
26:            $\sigma_t = \mathcal{D}_F(o_t)$
27:            $\sigma_{t+4} = \mathcal{D}_F(o_{t+4})$
28:            $\hat{c}_\ell = \mathcal{D}_R(\sigma_t, \sigma_{t+4})$
29:            $r = \hat{c}_\ell[0]$, the probability that $o_t, o_{t+4}$ were generated via expert rollouts.
30:            Use $r$ to train $\pi_\theta$ with via policy gradients (TRPO).
31:        **end for**
32:     **end for**
33: **end for**
34: **return** optimized policy $\pi_\theta$

---

## 6.1 ENVIRONMENTS

To evaluate our algorithm, we consider three environments in the MuJoCo physics simulator. There are two different versions of each environment, an expert variant and a novice variant. Our goal is to train a cost function that is domain agnostic, and hence can be trained with images on the expert domain but nevertheless produce a reasonable cost on the novice domain. See Figure 1 for a visualization of the differences between expert and novice environments for the three tasks.

**Point:** A pointmass attempts to reach a point in a plane. The color of the target and the camera angle change between domains.

**Reacher:** A two DOF arm attempts to reach a designated point in the plane. The camera angle, the length of the arms, and the color of the target point are changed between domains. Note that changing the camera angle significantly alters the image background color from largely gray to roughly 30 percent black. This presents a significant challenge for our method.

**Inverted Pendulum:** A classic RL task wherein a pendulum must be made to balance via control. For this domain, We only change the color of the pendulum and not the camera angle. Since there is no target point, we found that changing the camera angle left the domain invariant representations with too little information and resulted in a failure case. In contrast to some traditional renderings

of this problem, we do not terminate an episode when the agent falls but rather allow data collection to continue for a fixed horizon.

## 6.2   EVALUATIONS

***Is it possible to solve the third-person imitation learning problem in simple settings?*** In Figure 3, we see that our proposed algorithm is indeed able to recover reasonable policies for all three tasks we examined. Initially, the training is quite unstable due to the domain confusion wreaking havoc on the learned cost. However, after several iterations the policies eventually head towards reasonable local minima and the standard deviation over the reward distribution shrinks substantially. Finally, we note that the extracted feature representations used to complete this task are in fact domain-agnostic, as seen in Figure 9. Hence, the learning is properly taking place from a third-person perspective.

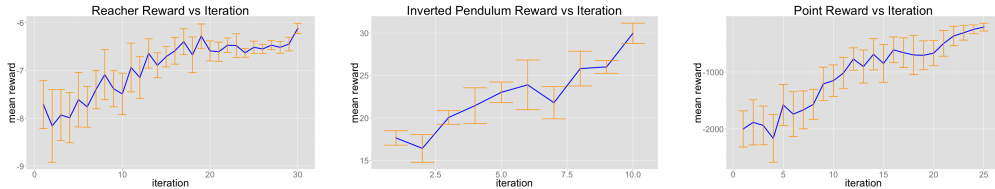

Figure 3: Reward vs training iteration for reacher, inverted pendulum, and point environments. The learning curves are averaged over 5 trials with error bars represent one standard deviation in the reward distribution at the given point.



Figure 4: Domain accuracy vs. training iteration for reacher, inverted pendulum, and point environments.

***Does the algorithm we propose benefit from both domain confusion and the multi-time step input?*** We answer this question with the experiments summarized in Figure 5. This experiment compares our approach with: (i) our approach without the domain confusion loss; (ii) our approach without the multi-time step input; (iii) our approach without the domain confusion loss and without the multi-time step input (which is very similar to the approach in Ho & Ermon (2016)). We see that adding domain confusion is essential for getting strong performance in all three experiments. Meanwhile, adding multi-time step input marginally improves the results. See also Figure 7 for an analysis of the effects of multi-time step input on the final results.

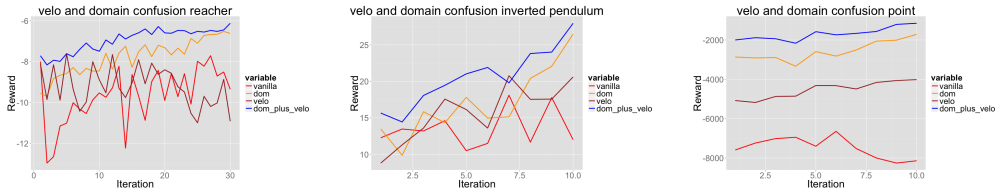

Figure 5: Reward vs iteration for reacher, inverted pendulum, and point environments with no domain confusion and no velocity (red), domain confusion (orange), velocity (brown), and both domain confusion and velocity (blue).

***How sensitive is our proposed algorithm to the selection of hyper-parameters used in deployment?***
Figure 6 shows the effect of the domain confusion coefficient $\lambda$, which trades off how much we should weight the domain confusion objective vs. the standard cost-recovery objective, on the final performance of the algorithm. Setting $\lambda$ too low results in slower learning and features that are not domain-invariant. Setting $\lambda$ too high results in an objective that is too quick to destroy information, which makes it impossible to recover an accurate cost.

For multi-time step input, one must choose the number of look-ahead frames that are utilized. If too small a window is chosen, the agent's actions have not affected a large amount of change in the environment and it is difficult to discern any additional class signal over static images. If too large a time-frame passes, causality becomes difficult to interpolate and the agent does worse than simply being trained on static frames. Figure 7 illustrates that no number of look-ahead frames is consistently optimal across tasks. However, a value of $4$ showed good performance over all tasks, and so this value was utilized in all other experiments.

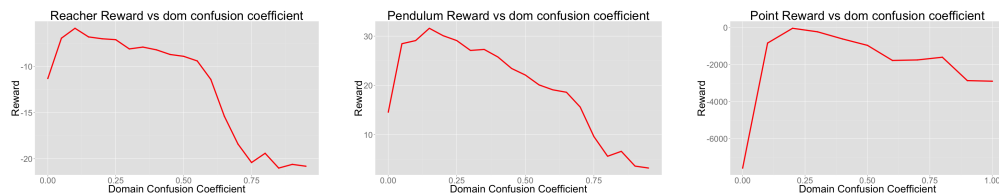

Figure 6: Reward of final trained policy vs domain confusion weight $\lambda$ for reacher, inverted pendulum, and point environments.



Figure 7: Reward of final trained policy vs number of look-ahead frames for reacher, inverted pendulum, and point environments.

***How sensitive is our algorithm to changes in camera angle?*** We present graphs for the reacher and point experiments wherein we exam the final reward obtained by a policy trained with third-person imitation learning vs the camera angle difference between the first-person and third-person perspective. We omit the inverted double pendulum experiment, as the color and not the camera angle changes in that setting and we found the case of slowly transitioning the color to be the definition of uninteresting science.

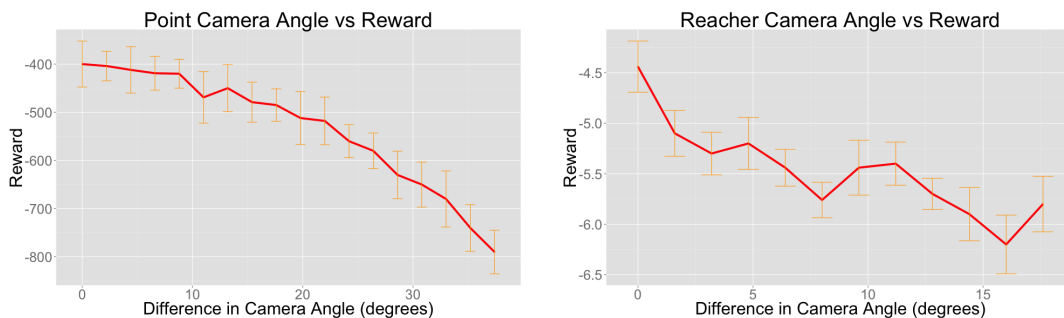

Figure 8: Point and reacher final reward after 20 epochs of third-person imitation learning vs the camera angle difference between the first and third-person perspective. We see that the point follows a fairly linear slope in regards to camera angle differences, whereas the reacher environment is more stochastic against these changes.

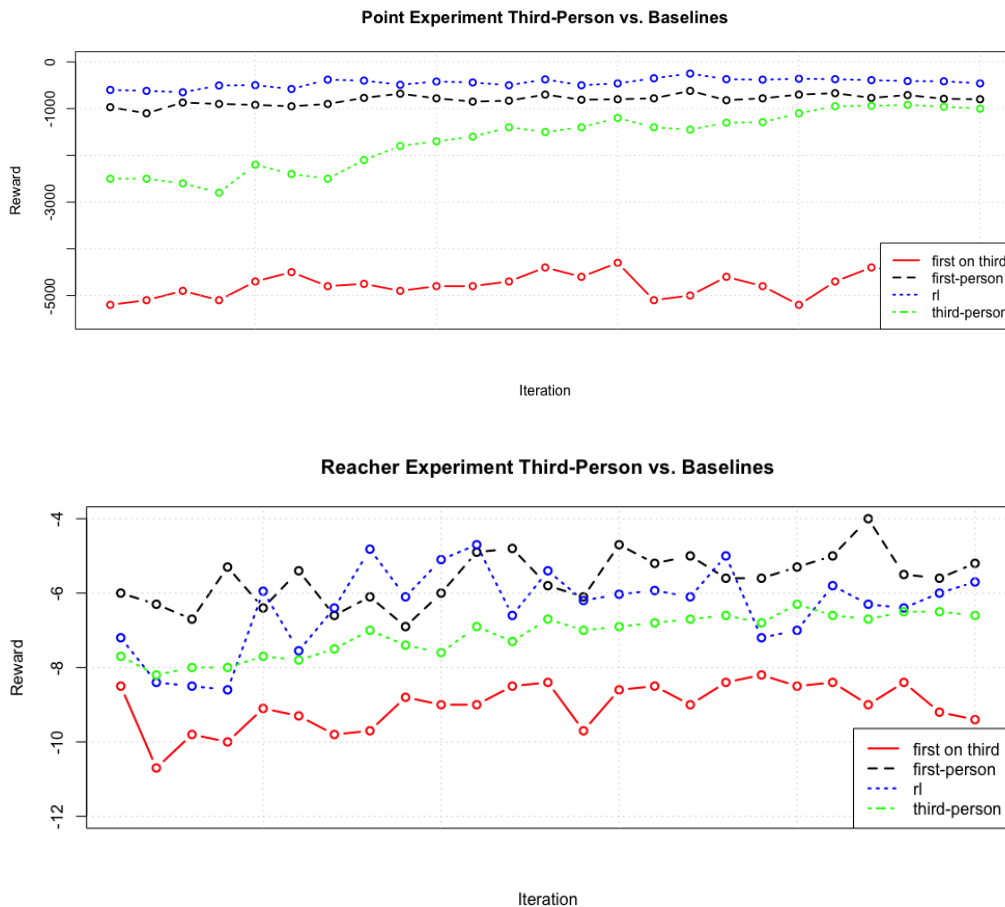

Figure 9: Learning curves for third-person imitation vs. three baselines: 1)RL with true reward, 2) first-person imitation, 3) attempting to use first-person features on the third-person agent.

***How does our method compare against reasonable baselines?*** We consider the following baselines for comparisons against third-person imitation learning. 1) Standard reinforcement learning with using full state information and the true reward signal. This agent is trained via TRPO. 2)

Standard GAIL (first-person imitation learning). Here, the agent receives first-person demonstration and attempts to imitate the correct behavior. This is an upper bound on how well we can expect to do, since we have the correct perspective. 3) Training a policy using first-person data and applying it to the third-person environment.

We compare all three of these baselines to third-person imitation learning. As we see in figure 9: 1) Standard RL, which (unlike the imitation learning approaches) has access to full state and true reward, helps calibrate performance of the other approaches. 2) First-person imitation learning is faced with a simpler imitation problem and accordingly outperforms third-person imitation, yet third-person imitation learning is nevertheless competitive. 3) Applying the first-person policy to the third-person agent fails miserably, illustrating that explicitly considering third-person imitation is important in these settings.

Somewhat unfortunately, the different reward function scales make it difficult to capture information on the variance of each learning curve. Consequently, in Appendix A we have included the full learning curves for these experiments with variance bars, each plotted with an appropriate scale to examine the variance of the individual curves.

## 7 DISCUSSION AND FUTURE WORK

In this paper, we presented the problem of third-person imitation learning. We argue that this problem will be important going forward, as techniques in reinforcement learning and generative adversarial learning improve and the cost of collecting first-person samples remains high. We presented an algorithm which builds on Generative Adversarial Imitation Learning and is capable of solving simple third-person imitation tasks.

One promising direction of future work in this area is to jointly train policy features and cost features at the pixel level, allowing the reuse of image features. Code to train a third person imitation learning agent on the domains from this paper is presented here: `https://github.com/bstadie/third_person_im`

## ACKNOWLEDGEMENTS

This work was done partially at OpenAI and partially at Berkeley. Work done at Berkeley was supported in part by Darpa under the Simplex program and the FunLoL program.

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

# 8 APPENDIX A: LEARNING CURVES FOR BASELINES

Here, we plot the learning curves for each of the baselines mentioned in the experiments section as a standalone plot. This allows one to better examine the variance of each individual learning curve.

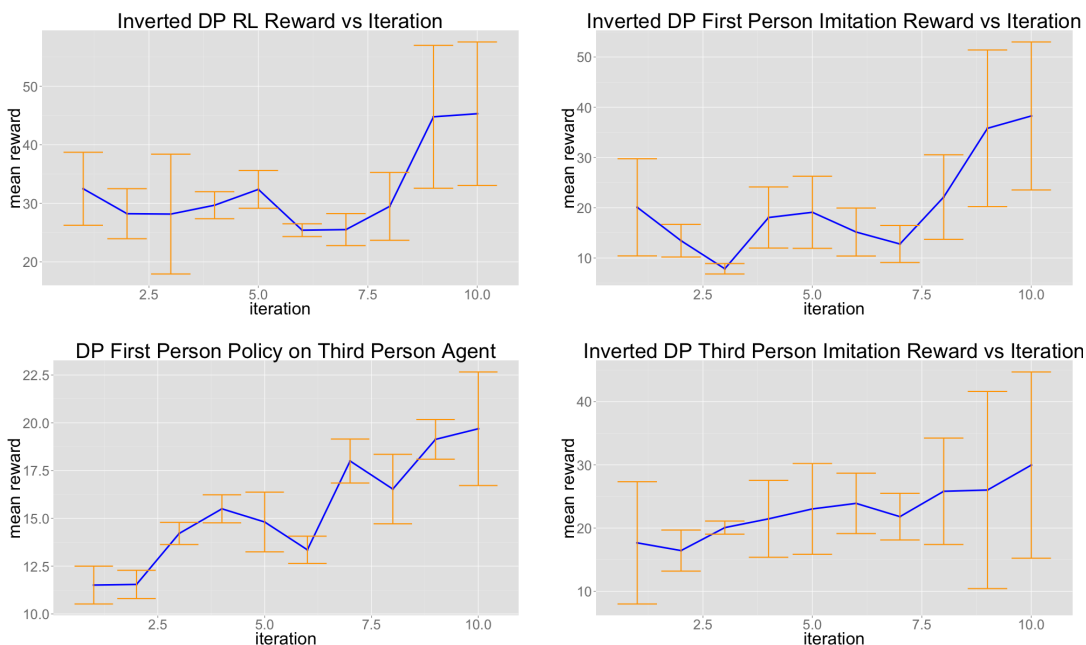

Figure 10: Inverted Pendulum performance under a policy trained on RL, first-person imitation learning, third-person imitation, and a first-person policy applied to a third-person agent.

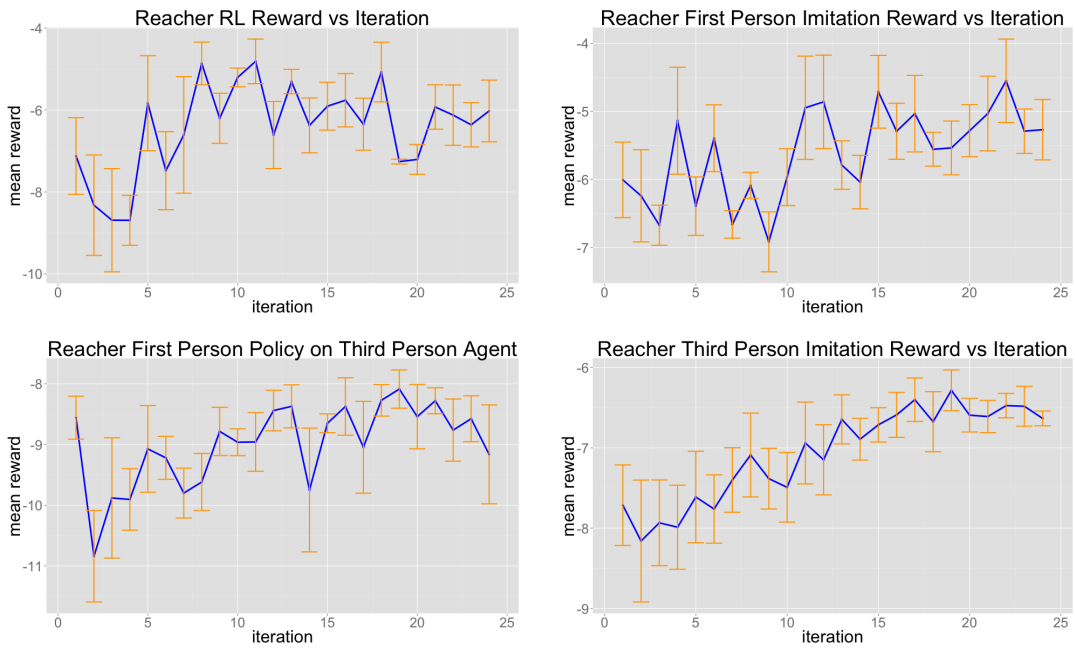

Figure 11: Reacher performance under a policy trained on RL, first-person imitation learning, third-person imitation, and a first-person policy applied to a third-person agent.

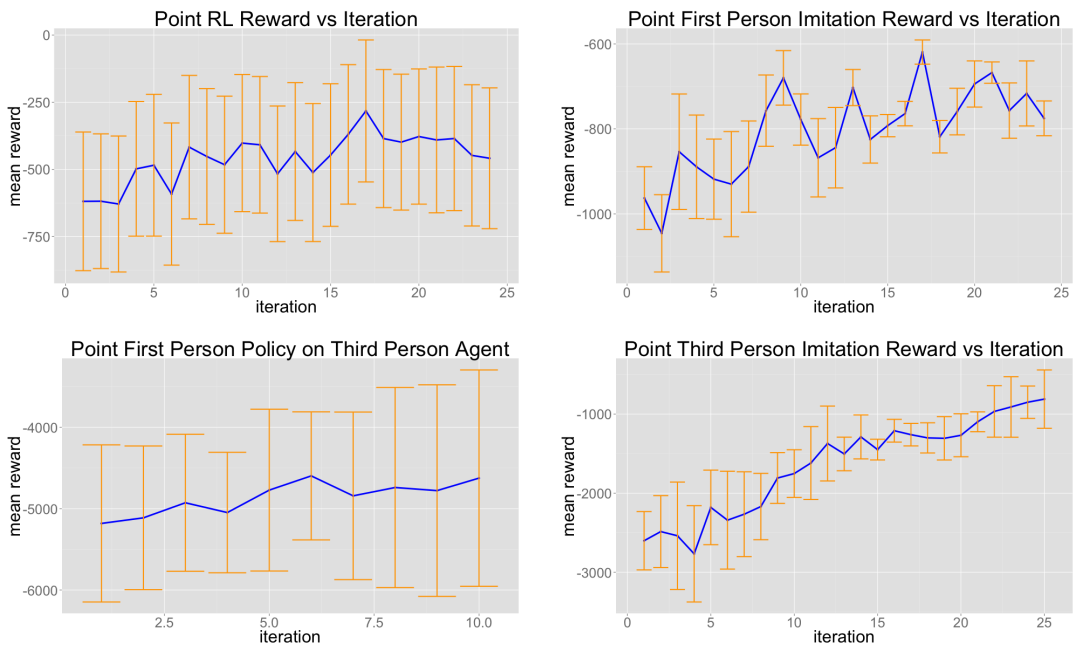

Figure 12: Point performance under a policy trained on RL, first-person imitation learning, third-person imitation, and a first-person policy applied to a third-person agent.

## 9 APPENDIX B: ARCHITECTURE PARAMETERS

Joint Feature Extractor: Input is images are size 50 x 50 with 3 channels, RGB. Layers are 2 convolutional layers each followed by a max pooling layer of size 2. Layers use 5 filters of size 3 each.

Domain Discriminator and the Class Discriminator: Input is domain agnostic output of convolutional layers. Layers are two feed forward layers of size 128 followed by a final feed forward layer of size 2 and a soft-max layer to get the log probabilities.

ADAM is used for discriminator training with a learning rate of 0.001. The RL generator uses the off-the-shelf TRPO implementation available in RLLab.

