# Peer review of "Third Person Imitation Learning"

_ICLR 2017 — accepted_

[Official Review · AnonReviewer3 · rating 5 · confidence 3 · 16 Dec 2016]
**Interesting idea but need more experiments**
soundness 4 · clarity 5

This paper proposed a novel adversarial framework to train a model from demonstrations in a third-person perspective, to perform the task in the first-person view. Here the adversarial training is used to extract a novice-expert (or third-person/first-person) independent feature so that the agent can use to perform the same policy in a different view point.

While the idea is quite elegant and novel (I enjoy reading it), more experiments are needed to justify the approach. Probably the most important issue is that there is no baseline, e.g., what if we train the model with the image from the same viewpoint? It should be better than the proposed approach but how close are they? How the performance changes when we gradually change the viewpoint from third-person to first-person? Another important question is that maybe the network just blindly remembers the policy, in this case, the extracted feature could be artifacts of the input image that implicitly counts the time tick in some way (and thus domain-agonistic), but can still perform reasonable policy. Since the experiments are conduct in a synthetic environment, this might happen. An easy check is to run the algorithm on multiple viewpoint and/or with blurred/differently rendered images, and/or with random initial conditions.

Other ablation analysis is also needed. For example, I am not fully convinced by the gradient flipping trick used in Eqn. 5, and in the experiments there is no ablation analysis for that (GAN/EM style training versus gradient flipping trick). For the experiments, Fig. 4,5,6 does not have error bars and is not very convincing.

[Official Review · AnonReviewer5 · rating 6 · confidence 4 · 16 Dec 2016 (modified: 20 Jan 2017)]
**Interesting idea for imitation learning. Paper could have been more general.**

The paper presents an interesting new problem setup for imitation learning: an agent tries to imitate a trajectory demonstrated by an expert but said trajectory is demonstrated in a different state or observation space than the one accessible by the agent (although the dynamics of the underlying MDP are shared). The paper proposes a solution strategy that combines recent work on domain confusion losses with a recent IRL method based on generative adversarial networks.

I believe the general problem to be relevant and agree with the authors that it results in a more natural formulation for imitation learning that might be more widely applicable.
There are however a few issues with the paper in its current state that make the paper fall short of being a great exploration of a novel idea. I will list these concerns in the following (in arbitrary order)
- The paper feels at times to be a bit hurriedly written (this also mainly manifests itself in the experiments, see comment below) and makes a few fairly strong claims in the introduction that in my opinion are not backed up by their approach. For example: "Advancements in this class of algorithms would significantly improve the state of robotics, because it will enable anyone to easily teach robots new skills"; given that the current method to my understanding has the same issues that come with standard GAN training (e.g. instability etc.) and requires a very accurate simulator to work well (since TRPO will require a large number of simulated trajectories in each step) this seems like an overstatement.
  There are some sentences that are ungrammatical or switch tense in the middle of the sentence making the paper harder to read than necessary, e.g. Page 2: "we find that this simple approach has been able to solve the problems"
- The general idea of third person imitation learning is nice, clear and (at least to my understanding) also novel. However, instead of exploring how to generally adapt current IRL algorithms to this setting the authors pick a specific approach that they find promising (using GANs for IRL) and extend it. A significant amount of time is then spent on explaining why current IRL algorithms will fail in the third-person setting. I fail to see why the situation for the GAN based approach is any different than that of any other existing IRL algorithm. To be more clear: I see no reason why e.g. behavioral cloning could not be extended with a domain confusion loss in exactly the same way as the approach presented. To this end it would have been nice to rather discuss which algorithms can be adapted in the same way (and also test them) and which ones cannot. One straightforward approach to apply any IRL algorithm would for example be to train two autoencoders for both domains that share higher layers + a domain confusion loss on the highest layer, should that not result in features that are directly usable? If not, why?
- While the general argument that existing IRL algorithms will fail in the proposed setting seems reasonable it is still unfortunate that no attempts have been made to validate this empirically. No comparison is made regarding what happens when one e.g. performs supervised learning (behavioral cloning) using the expert observations and then transfers to the changed domain. How well would this work in practice ? Also, how fast can different IRL algorithms solve the target task in general (assuming a first person perspective) ?
- Although I like the idea of presenting the experiments as being directed towards answering a specific set of questions I feel like the posed questions somewhat distract from the main theme of the paper. Question 2 suddenly makes the use of additional velocity information to be a main point of importance and the experiments regarding Question 3 in the end only contain evaluations regarding two hyperparameters (ignoring all other parameters such as the parameters for TRPO, the number of rollouts per iteration, the number of presented expert episodes and  the design choices for the GAN). I understand that not all of these can be evaluated thoroughly in a conference paper but I feel like some more experiments or at least some discussion would have helped here.
- The presented experimental evaluation somewhat hides the cost of TRPO training with the obtained reward function. How many roll-outs are necessary in each step?
- The experiments lack some details: How are the expert trajectories obtained? The domains for the pendulum experiment seem identical except for coloring of the pole, is that correct (I am surprised this small change seems to have such a detrimental effect)? Figure 3 shows average performance over 5 trials, what about Figure 5 (if this is also average performance, what is the variance here)? Given that GANs are not easy to train, how often does the training fail/were you able to re-use the hyperparameters across all experiments?

UPDATE:
I updated the score. Please see my response to the rebuttal below.

[Official Review · AnonReviewer2 · rating 6 · confidence 4 · 16 Dec 2016]
originality 1

The paper extends the imitation learning paradigm to the case where the demonstrator and learner have different points of view. This is an important contribution, with several good applications.  The main insight is to use adversarial training to learn a policy that is robust to this difference in perspective.  This problem formulation is quite novel compared to the standard imitation learning literature (usually first-order perspective), though has close links to the literature on transfer learning (as explained in Sec.2).

The basic approach is clearly explained, and follows quite readily from recent literature on imitation learning and adversarial training.

I would have expected to see comparison to the following methods added to Figure 3:
1)  Standard 1st person imitation learning using agent A data, and apply the policy on agent A.  This is an upper-bound on how well you can expect to do, since you have the correct perspective.
2)  Standard 1st person imitation learning using agent A data, then apply the policy on agent B.  Here, I expect it might do less well than 3rd person learning, but worth checking to be sure, and showing what is the gap in performance.
3)  Reinforcement learning using agent A data, and apply the policy on agent A.  I expect this might do better than 3rd person imitation learning but it might depend on the scenario (e.g. difficulty of imitation vs exploration; how different are the points of view between the agents). I understand this is how the expert data is collected for the demonstrator, but I don’t see the performance results from just using this procedure on the learner (to compare to Fig.3 results).

Including these results would in my view significantly enhance the impact of the paper.

[Final Decision · Program Chairs · 06 Feb 2017]
**ICLR committee final decision**

pros:
 - new problem
 - huge number of experimental evaluations, based in part on open-review comments
 
 cons:
 - the main critiques related to not enough experiments being run; this has been addressed in the revised version
 
 The current reviewer scores do not yet reflect the many updates provided by the authors.
 I therefore currently learn in favour of seeing this paper accepted.